# Genomic Prostate Score: A New Tool to Assess Prognosis and Optimize Radiation Therapy Volumes and ADT in Intermediate-Risk Prostate Cancer

**DOI:** 10.3390/cancers15030945

**Published:** 2023-02-02

**Authors:** Yazid Belkacemi, Kamel Debbi, Gabriele Coraggio, Jérome Bendavid, Maya Nourieh, Nhu Hanh To, Mohamed Aziz Cherif, Carolina Saldana, Alexandre Ingels, Alexandre De La Taille, Gokoulakrichenane Loganadane

**Affiliations:** 1Department of Radiation Oncology and Henri Mondor Breast Center, Henri Mondor Hospital, APHP, University of Paris Est Créteil (UPEC), IMRB, INSERM U 955, 94000 Créteil, France; 2Department of Pathology, Henri Mondor Hospital, University of Paris Est Créteil (UPEC), IMRB, INSERM U 955, 94000 Créteil, France; 3Department of Medical Oncology, Henri Mondor Hospital, University of Paris Est Créteil (UPEC), 94000 Créteil, France; 4Department of Urology, Henri Mondor Hospital, University of Paris Est Créteil (UPEC), 94000 Créteil, France

**Keywords:** prostate cancer, gleason score, radiation therapy, genomic prostate score, risk assessment

## Abstract

**Simple Summary:**

Current standard clinical risk-stratification systems do not sufficiently reflect the disease heterogeneity for prostate cancer. Intermediate risk prostate cancer represents a highly heterogeneous risk group with different treatment options available such as surgery, radiation therapy and hormone therapy. Genomic Prostate Score which can obtained from prostate core biopsies could help to personalize treatment for men with intermediate-risk prostate cancer. The main aim of this present prospective study was to assess the impact of Genomic Prostate Score when compared to clinical risk factors alone in this population. This research study demonstrated that use of Genomic Prostate Score in a series of 30 patients with intermediate risk prostate cancer resulted in major shift in risk groups in most patients. This genomic score represents a potential impactful tool for treatment decision and better personalization of treatment for patients with intermediate risk in daily practice.

**Abstract:**

Genomic classifiers such as the Genomic Prostate Score (GPS) could help to personalize treatment for men with intermediate-risk prostate cancer (I-PCa). In this study, we aimed to evaluate the ability of the GPS to change therapeutic decision making in I-PCa. Only patients in the intermediate NCCN risk group with Gleason score 3 + 4 were considered. The primary objective was to assess the impact of the GPS on risk stratification: NCCN clinical and genomic risk versus NCCN clinical risk stratification alone. We also analyzed the predictive role of the GPS for locally advanced disease (≥pT3+) and the potential change in treatment strategy. Thirty patients were tested for their GPS between November 2018 and March 2020, with the median age being 70 (45–79). Twenty-three patients had a clinical T1 stage. Eighteen patients were classified as favorable intermediate risk (FIR) based on the NCCN criteria. The median GPS score was 39 (17–70). Among the 23 patients who underwent a radical prostatectomy, Gleason score 3 + 4 was found in 18 patients. There was a significant correlation between the GPS and the percentage of a Gleason grade 4 or higher pattern in the surgical sample: correlation coefficient r = 0.56; 95% CI = 0.2–0.8; *p* = 0.005. In this study, the GPS combined with NCCN clinical risk factors resulted in significant changes in risk group.

## 1. Introduction

Prostate cancer (PCa) is the second-most commonly diagnosed cancer and the sixth leading cause of cancer death among men worldwide, with an estimated 1,276,000 new cases and 359,000 deaths in 2018 [1]. The global PCa burden is expected to rise to almost 2.3 million new cases and 740,000 deaths by 2040 simply due to the growth and aging of the population.

Risk stratification and treatment decision making for localized PCa traditionally relies on a combination of clinical staging, PSA level, and biopsy results (Gleason score) [2,3].

Intermediate-risk PCa consists of a highly heterogeneous population with different treatment options. Evidence shows that favorable intermediate risk (FIR) PCa patients have cancer-specific mortality rates similar to those with low-risk PCa and, thus, may be candidates for active surveillance, dose-escalated radiation therapy (RT) without short-term androgen deprivation therapy (ADT). Conversely, patients with unfavorable intermediate-risk (UIR) PCa have prostate cancer-specific mortality rates similar to those reported in high-risk patients. Theoretically, they would require a more aggressive multimodal approach, including long-term ADT in addition to standard-dose or dose-escalated radiation therapy with a brachytherapy boost, for example.

In a review that focused on intermediate-risk PCa, Zumsteg and Zelefsky presented this entity as a highly heterogenous one. This clinical heterogeneity makes application of a uniform treatment standard difficult, especially knowing that more than 50% of variability in cure rates for PCa is related to clinical factors other than tumor stage, PSA level, and Gleason score, such as the proportion of positive biopsy cores, perineural invasion, pretreatment PSA velocity, and primary Gleason pattern [4,5,6,7].

In this context, the liquid biopsy field in PCa has advanced exponentially, developing prognostic and predictive biomarkers including circulating tumor cells, extracellular vesicles, circulating tumor DNA, RNA, and holding promise for a minimally invasive approach of monitoring tumor response [8].

Similarly, genomic classifiers (GC) have been shown to be promising for the identification of aggressive PCa from tumoral tissue and for guiding treatment decisions with different commercially available profiling panels including Prolaris^®^, ProMark^®^, Oncotype DX^®^, and Decipher^®^ [9]. The bulk of the evidence for the GC is available for improving risk stratification in the postoperative setting and for guidance for adjuvant radiation therapy. Less data exists for patients treated with definitive radiation therapy, but GC might facilitate personalized oncologic treatments in various perspectives in all disease stages.

In this study, we aimed to evaluate the ability of the Genomic Prostate Score (GPS) for potentially modifying treatment decision in intermediate-risk PCa in terms of indication and duration of ADT and volumes of radiation compared to clinical parameters alone.

## 2. Materials and Methods

### 2.1. Patients and Study Design

The charts of the patients who underwent diagnostic prostatic core biopsies were all reviewed in the weekly genitourinary multidisciplinary tumor board, with eligible patients for the GPS study prospectively recruited between November 2018 and March 2020. Inclusion criteria were: intermediate NCCN (National Comprehensive Cancer Network) clinical risk group and Gleason grade 2 (Gleason score 3 + 4). Exclusion criteria were: high-risk NCCN features including PSA > 20, clinical T3, or >Gleason grade 2, or the presence of clinical nodal or distant metastases. Other exclusion criteria were: age > 80, prior prostate core biopsies, ongoing active surveillance program for PCa, prior ADT or pelvic irradiation, other active cancer, or any other life-threatening disease. The presence of prostatic intra-epithelial neoplasia was also an exclusion criterion, as this histologic pattern was excluded from GPS studies. GPS score tests were proposed to all eligible patients. The first 30 positive responders who completed and returned the approval form were included in the analysis.Multiparametric prostate MRI, bone scan, and CT scan were systematically performed. Choline PET CT or PET MRI were optional for the detection of distant metastases. GPS tests were provided free of charge by Genomic Health based on the agreement for the Physician Experience Program (PEP). The purpose of the PEP is to allow treating physicians to gain experience and acquaint themselves with the Oncotype DX^®^ Genomic Prostate Score. Treatments were provided blind to the GPS results. The study was conducted in accordance with the Declaration of Helsinki and was approved by the HOPITAL DE MONDOR Institutional Review Board (project number 16169, N° ID-RCB: 2016-A00789-42, 22 September 2016).

### 2.2. Biopsy and Tumor Selection Process

All patients underwent transrectal ultrasound-guided 18-core biopsies and other additional MRI-targeted biopsies were performed in the case of identifiable lesions. Formalin-fixed, paraffin-embedded specimens obtained from biopsy tissue were reviewed by the pathologist according to the 2016 International Society of Urological Pathology consensus guidelines. Tissues were selected for analysis with the GPS assay from the biopsy core associated with the largest tumor amount with Gleason score 4. Eight unstained slides of 5 µm were retrieved from the most predominant core biopsy with Gleason score 4.

Samples underwent standard, pre-established quality control measures prior to generating the GPS result. Samples were referred to the central laboratory of Genomic Health (Redwood City, CA, USA) using a specimen transportation kit.

### 2.3. GPS Assay Description

The Oncotype DX^®^ Genomic Prostate Score test is a quantitative reverse transcriptase polymerase chain reaction (RT-PCR) assay that measures the expression levels of 17 genes (12 cancer-related and 5 housekeeper) in Messenger RNA extracted from microdissected tumor tissue obtained from fixed prostate core biopsies. It provides a score scaled from 0 to 100 as a molecular measure of increasing tumor aggressiveness and is correlated with the risk of finding an adverse pathology feature (Gleason = 4 + 3 and/or pT3+) on the radical prostatectomy specimen [10]. A global risk group was provided based on the combination of clinical factors and the genomic score (GPS). Evidence shows that the GPS is also predictive of the biochemical recurrence and of the risk of metastases [11]. The estimated risk of metastases and prostate cancer death at 10 years was provided, along with the GPS results estimated on clinical validation of the GPS based on radical prostatectomy. 

### 2.4. Endpoints

The primary objective was to assess the impact of the GPS on risk stratification: NCCN clinical and genomic risk stratification versus NCCN clinical risk stratification alone. In the surgical cohort, we aimed to analyze the correlation between the GPS and the percentage of a Gleason score 4 higher pattern in the surgical sample using a Pearson correlation test. We also analyzed the predictive role of the GPS for locally advanced disease (=pT3+) using logistic regression. Statistical significance was defined as *p* < 0.05. Irrespective of the treatment actually offered to the patient, radiation was considered as the definitive treatment and the use of ADT, its duration, and the radiation volumes were recommended by radiation oncologists based on the clinical factors alone (PSA level, clinical stage, percentage of positive biopsies, NCCN risk group) and then including the GPS. The follow-up was estimated between the date of surgery or the first day of the radiation and the last dedicated follow-up consultation. All analyses were performed using R software version 3.5.1 (R project, Vienna, Austria).

## 3. Results

### 3.1. Patients’ Selection Process

Between November 2018 and March 2020, 164 patients who underwent prostate core biopsies were identified with Gleason score 3 + 4. Among them, 44 were excluded because of the presence of one of the exclusion criteria (prior prostate core biopsies, age > 80, active cancer, or any other life-threatening disease). The remaining 120 patients were considered eligible and the 30 first responders were selected for the GPS test.

### 3.2. Patients Characteristics

Among the 30 patients who were tested for the GPS, the median age was 70 (45–79) with a median PSA level of 7 ng/mL (0.9–16) (Table 1). Of the 30 patients, 23 had a clinical T1 stage and 7 had a clinical T2 stage. There were 12 patients who had ≥ 50% of their core biopsies involved, while 18 had < 50% of their core biopsies involved. Surgery was performed in 23 patients and 7 patients underwent definitive radiation combined with a short-term ADT.

### 3.3. Risk Stratification

A total of 18 patients were classified as favorable intermediate risk (FIR) and 12 patients were classified as unfavorable intermediate risk (UIR), as per the NCCN clinical stratification. The median GPS score based on the gene expression was 39 (17–70). The risk category based on the combination of NCCN clinical parameters and the genomic features was provided by Genomic Health for the 30 patients included in the study. Reclassification was observed in 66% of cases (20/30), with 60% (18/30) associated with a higher risk category and 6% (2/30) associated with a lower risk category. The potential impact on treatment decision based on the consensual opinion of radiation oncologists was also significant, with 80% (24/30) of change consisting of intensification in 77% (23/30) and de-escalation in 3% (1/30). Furthermore, 6 patients were reclassified as FIR, 13 as UIR, and 11 as high risk (the details are presented in Figure 1). After a median follow-up of 11 months, 2 patients experienced biochemical recurrence after radical prostatectomy and underwent salvage radiation associated with ADT.

### 3.4. Correlation between GPS and Histological Features

Among the 23 patients who underwent radical prostatectomy, Gleason score 3 + 4 was found in 18 patients and Gleason score 4 + 3 was found in 5 patients. The Pearson test showed a significant correlation between the GPS and the percentage of a Gleason grade 4 or higher pattern in the surgical sample: correlation coefficient r = 0.56; 95% CI = 0.2–0.8; *p* = 0.005. Among the 23 patients who underwent radical prostatectomy, 11 were found to have locally advanced disease in the surgical specimen (=pT3+) (pT3a: 10 cases, pT3b: 1 case) and 12 had a localized disease (pT2). The GPS was not significantly predictive of the locally advanced disease (=pT3+) in the radical prostatectomy sample with the logistic regression model: Odds ratio = 1.07; 95% CI = 0.99–1.14; *p* = 0.09. Among the 23 patients who underwent radical prostatectomy, 20 patients had a lymph node dissection. One patient was found to have a single positive lymph node metastasis among the 10 lymph nodes removed (1N+/10N).

### 3.5. GPS IMPACT on Therapeutic Decision Making and RT Volumes

The potential impact of the GPS in treatment decision is presented in Table 2. Among the 30 patients included in the study, 23 would have received treatment intensification (77%), 1 would have received treatment de-escalation (3%), and 6 would have had the treatment unchanged.

## 4. Discussion

The management of localized PCa is currently based on clinical risk groups. In the era of personalized medicine, several prognostic GC such as Oncotype DX GPS^®^, Prolaris^®^, ProMark^®^, DNA-ploidy, and Decipher^®^ have been investigated in PCa [12]. The investigation of GC for guiding radiation therapy as the primary treatment has been limited, but several ongoing prospective trials are addressing this question (Table 3). Genomic classifiers may provide greater insight into PCa tumor biology and more accurately stratify PCa risk for adverse outcomes prior to definitive radiation therapy. Few retrospective studies investigated the clinical utility of these genomic signatures based on biopsy samples in patients with localized PCa treated with definitive radiotherapy.

Decipher^®^ is a genomic classifier based on the expression of 22 selected RNA markers using formalin-fixed, paraffin-embedded tissue. In the context of definitive radiotherapy for PCa, Decipher^®^ was investigated on biopsy specimens from the NRG biobank from patients enrolled in the NRG/RTOG 9202, 9413, and 9902 phase III randomized radiation therapy trials [13]. With a median follow-up of 11 years, the Decipher score was shown to be a significant prognostic factor for distant metastases, PCa-specific mortality, and overall survival in univariate and multivariate analysis in a high-risk PCa population.

In a prospective study, Berlin et al. analyzed outcomes of 121 patients with NCCN intermediate-risk PCa treated with definitive RT without ADT. Decipher^®^ outperformed all other indices in its prediction of distant metastases (HR = 2.05; 95% CI = 1.24–4.24) [14].

On the other hand, Oncotype DX GPS^®^ is a 17-gene RT-PCR-based assay that evaluates the expression of 12 cancer-related genes and 5 housekeeper genes from prostate core biopsies. Oncotype DX GPS^®^ assay generates a genomic score from 0 to 100, with higher scores indicative of more aggressive disease. Oncotype DX GPS^®^ is recognized as an option for patients with very low and low-risk localized PCa who are candidates for active surveillance according to National Comprehensive Cancer Network (NCCN) and American Society for Clinical Oncology guidelines [12].

Oncotype DX GPS^®^ has also been validated as an independent prognostic factor of adverse pathology, biochemical recurrence, distant metastasis, and prostate cancer-related death in men with localized PCa after radical prostatectomy in NCCN very low, low, and favorable intermediate risk population [10,11]. In a higher risk cohort (UIR patients), Cullen et al. reported that the GPS score was a strong independent predictor of recurrence, metastatic evolution, and death [15]. In that study, GPS > 40 was associated with worse outcomes and may require treatment intensification.

A recently published retrospective study reported outcomes of 238 men with localized PCa treated with radiation therapy in North Carolina from 2000 to 2016. The NCCN very low or low, favorable intermediate, unfavorable intermediate, high and very-high-risk patients represented 8%, 23%, 37%, 21%, and 11%, respectively.

With a median follow-up of 7.6 years. GPS results per 20-unit increase were significantly associated with biochemical failure (HR = 3.62; 95% CI = 2.59–5.02), distant metastases (HR = 4.48; 95% CI = 2.75–7.38), and PCa death (HR = 5.36; 95% CI = 3.06–9.76) in univariable analysis. Moreover, GPS results retained statistical significance in the multivariable analysis [16].

Interestingly, the dichotomization of the GPS with a cut-off of 40 (0–40 vs. 41–100) yielded similar results, with patients with GPS > 40 predictive of worse outcomes. 

To the best of our knowledge, our study is the first to focus on intermediate NCCN risk group patients with Gleason grade 2 (Gleason score 3 + 4). Our aim was to assess the potential clinical impact of the GPS on risk stratification (NCCN clinical and genomic risk stratification versus NCCN clinical risk stratification alone). The risk group migration occurred in 66% of patients, with 60% associated with a higher risk category.

The potential impact on treatment decision based on the consensual opinion of radiation oncologists was even more significant, with 80% of change consisting mainly of intensification in 77% (23/30). The median GPS score in our study was 39, ranging from 17 to 70. Interestingly, this level is close to the threshold of 40 from the surgical cohort of Cullen et al. and the radiation cohort associated with poor outcomes [15,16].

Intensification strategies for intermediate unfavorable and high risk include ADT, prophylactic pelvic nodal irradiation, and brachytherapy boost.

In intermediate-risk PCa, the use of ADT is controversial with modern doses of radiation since its benefit is limited to biochemical control with overall survival [4].

Long-term ADT is known to improve disease-free survival and overall survival with a superiority of long over short term for high-risk patients [17]. In a recent phase 3 trial, prophylactic pelvic nodal irradiation improved biochemical free survival and disease survival in high-risk PCa [18].

To our knowledge, this report represents the first prospective study investigating the potential clinical utility of Oncotype DX GPS^®^ in patients with PCa with Gleason score 3 + 4 based on 18-core biopsies. The correlation between the GPS score and the percentage of Gleason grade 4 or higher pattern in the surgical specimen was consistent with prior studies. Major limitations related to the study need to be acknowledged. First, monocentric and observational design, and the small size of the cohort (*n* = 30), represent notable limitations. Second, recommendations related to ADT and radiation volumes were left to the appreciation of the comity of radiation oncologists based on clinical parameters (PSA level, clinical stage, percentage of positive biopsies, NCCN risk group) and genomic score. Different therapies may be offered for two patients within the same NCCN risk group and some of the treatment suggestions were not in accordance with NCCN guidelines. Third, the use of brachytherapy boost could also have been the part of the treatment suggestion. Finally, the follow-up was insufficient (11 months) to study the correlation between GPS and clinical outcomes. 

## 5. Conclusions

Intermediate-risk PCa represents a highly heterogeneous group with different treatment options available. In this study, the GPS combined with NCCN clinical risk factors resulted in significant changes in risk group definition. From a radiation oncologist point of view, this may have resulted in significant differences in terms of radiation volumes and duration of ADT. Future prospective studies exploring genomic classifiers to further personalize therapy in intermediate-risk PCa should be performed.

## Figures and Tables

**Figure 1 cancers-15-00945-f001:**
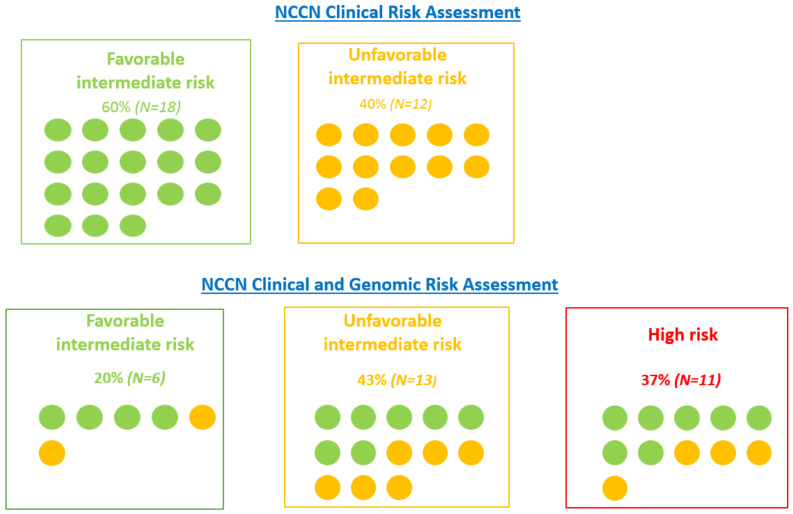
Clinical and genomic risk stratification. Classification based on NCCN clinical risk group alone and combined NCCN clinical and genomic risk group (GPS). The combined risk group (based on NCCN risk group and genomic score) was provided in the report by Genomic Health. Each round represents a patient.

**Table 1 cancers-15-00945-t001:** Patients’ characteristics.

Characteristics	*N* = 30	%
**Age (years)**		
Median (range)	70 (45–79)	-
**PSA (ng/mL)**		
Median (range)	7 (0.9–16)	-
**Clinical T Stage**		
T1	23	77
T2	7	23
**Prostate Volume (mL)**		
Median (range)	40 (15–75)	-
**Number of Positive Cores**		
Median (range)	7.5 (1–17)	-
**Number of Cores Sampled**		
Median (range)	18 (18–22)	-
**Positive Cores/Cores Sampled Ratio**		
<50%	18	60
≥50%	12	40
**NCCN Risk Group**		
Intermediate favorable	18	60
Intermediate unfavorable	12	40
**Upfront Treatment**		
Surgery	23	77
Radiation therapy	7	23

**Table 2 cancers-15-00945-t002:** Potential impact of GPS on treatment decision.

Patient Number	NCCN Clinical Risk Group	Decision Radiation + Hormonal Therapy (Based on Clinical Factors)	GPS	NCCN + GPS Risk Group	Estimated Metastatic Risk at 10 Years (%) Based on GPS Validation Studies	Estimated Prostate Cancer Death at 10 Years (%) Based on GPS Validation Studies	Decision Radiation Based on NCCN + GPS	Impact of GPS
1	Unfavorable intermediate	RT prostate alone with ADT (short term)	17	Favorable intermediate	4	1	RT prostate alone without ADT	De-escalation
2	Favorable intermediate	RT prostate alone with ADT (short term)	53	High	17	3	RT of the pelvic nodes and the prostate with ADT (long term)	Intensification
3	Favorable intermediate	RT prostate alone without ADT	33	Unfavorable intermediate	7	1	RT of the pelvic nodes and the prostate with ADT (short term)	Intensification
4	Unfavorable intermediate	RT prostate alone without ADT	37	Unfavorable intermediate	9	1	RT of the pelvic nodes and the prostate with ADT (short term)	Intensification
5	Unfavorable intermediate	RT prostate alone without ADT	19	Favorable intermediate	4	1	RT prostate alone without ADT	No change
6	Favorable intermediate	RT prostate alone with ADT (short term)	39	Unfavorable intermediate	10	1	RT of the pelvic nodes and the prostate with ADT (short term)	Intensification
7	Favorable intermediate	RT prostate alone without ADT	40	Unfavorable intermediate	10	1	RT of the pelvic nodes and the prostate with ADT (short term)	Intensification
8	Favorable intermediate	RT prostate alone without ADT	30	Unfavorable intermediate	7	1	RT of the pelvic nodes and the prostate with ADT (short term)	Intensification
9	Favorable intermediate	RT prostate alone without ADT	46	High	13	2	RT of the pelvic nodes and the prostate with ADT (long term)	Intensification
10	Favorable intermediate	RT prostate alone without ADT	70	High	23	4	RT of the pelvic nodes and the prostate with ADT (long term)	Intensification
11	Unfavorable intermediate	RT prostate alone without ADT	25	Unfavorable intermediate	5	1	RT of the pelvic nodes and the prostate with ADT (short term)	Intensification
12	Favorable intermediate	RT prostate alone without ADT	47	High	13	2	RT of the pelvic nodes and the prostate with ADT (long term)	Intensification
13	Favorable intermediate	RT prostate alone without ADT	36	Unfavorable intermediate	8	1	RT prostate alone without ADT	Intensification
14	Unfavorable intermediate	RT prostate alone with ADT (short term)	39	Unfavorable intermediate	10	1	RT of the pelvic nodes and the prostate with ADT (short term)	Intensification
15	Unfavorable intermediate	RT prostate alone without ADT	42	High	11	1	RT of the pelvic nodes and the prostate with ADT (long term)	Intensification
16	Favorable intermediate	RT prostate alone without ADT	19	Favorable intermediate	4	1	RT prostate alone without ADT	No change
17	Favorable intermediate	RT prostate alone without ADT	18	Favorable intermediate	4	1	RT prostate alone without ADT	No change
18	Favorable intermediate	RT prostate alone without ADT	18	Favorable intermediate	4	1	RT prostate alone without ADT	No change
19	Unfavorable intermediate	RT prostate alone without ADT	40	Unfavorable intermediate	10	1	RT of the pelvic nodes and the prostate with ADT (short term)	Intensification
20	Favorable intermediate	RT prostate alone without ADT	56	High	19	3	RT of the pelvic nodes and the prostate with ADT (long term)	Intensification
21	Favorable intermediate	RT prostate alone without ADT	26	Unfavorable intermediate	6	1	RT of the pelvic nodes and the prostate with ADT (short term)	Intensification
22	Unfavorable intermediate	RT prostate alone without ADT	20	Unfavorable intermediate	4	1	RT prostate alone with ADT (short term)	Intensification
23	Favorable intermediate	RT prostate alone without ADT	18	Favorable intermediate	4	1	RT prostate alone without ADT	no change
24	Favorable intermediate	RT prostate alone without ADT	34	Unfavorable intermediate	8	1	RT of the pelvic nodes and the prostate with ADT (short term)	Intensification
25	Unfavorable intermediate	RT prostate alone with ADT (short term)	43	High	11	1	RT of the pelvic nodes and the prostate with ADT (long term)	Intensification
26	Unfavorable intermediate	RT prostate alone without ADT	58	High	20	3	RT of the pelvic nodes and the prostate with ADT (long term)	Intensification
27	Favorable intermediate	RT prostate alone without ADT	47	High	12	2	RT of the pelvic nodes and the prostate with ADT (long term)	Intensification
28	Unfavorable intermediate	RT prostate alone with ADT (short term)	55	High	18	3	RT of the pelvic nodes and the prostate with ADT (long term)	Intensification
29	Favorable intermediate	RT prostate alone without ADT	53	High	17	3	RT of the pelvic nodes and the prostate with ADT (long term)	Intensification
30	Unfavorable intermediate	RT of the pelvic nodes and the prostate with ADT (short term)	39	Unfavorable intermediate	10	1	RT of the pelvic nodes and the prostate with ADT (short term)	No change

**Table 3 cancers-15-00945-t003:** Selected ongoing trials evaluating genomic biomarkers to guide treatment decisions in patients undergoing definitive radiation therapy.

Trial Name	Full Name of Trial	Common Name	Phase	Participants (Number)	Status (July 2022)
**NRG GU009** **NCT04513717**	Two studies for patients with high-risk prostate cancer testing less intense treatment for patients with a low gene risk score and testing a more intense treatment with a high gene risk score.	PREDICT-RT	III	2478	Recruiting
**NRG GU010** **NCT05050084**	Two studies for patients with unfavorable intermediate-risk prostate cancer testing less intense treatment for patients with a low gene risk score and testing a more intense treatment with a high gene risk score.	GUIDANCE	III	2050	Recruiting
**NCT04396808**	Genomics in Michigan to adjust outcomes in prostate cancer for men with newly diagnosed favorable-risk prostate cancer.	G-MAJOR	III	350	Recruiting

## Data Availability

The authors are in possession of the datasets related to the study and are willing to share with the editorial board if needed.

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
