# Peer review of "Genomic Prostate Score: A New Tool to Assess Prognosis and Optimize Radiation Therapy Volumes and ADT in Intermediate-Risk Prostate Cancer"

_cancers, 2023, doi:10.3390/cancers15030945_

Round 1

Reviewer 1 Report

The manuscript explored the utility of adding Oncotype DX GPS in treatment decision making for NCCN intermediate risk prostate cancer. The following comments should be addressed before the manuscript is considered for publication. 

Major comments:

1. Figure 1, the authors showed that the addition of GPS changed NCCN risk stratification. Please elaborate in detail how GPS changed patient's risk group since this is not described in the NCCN guideline.

2. Table 2 described treatment decision based on risk stratification. It is intriguing to notice that for patients of the same risk group, different treatment regimens were offered (eg. patient 2 and 3). Please provide more information on how the treatment regimen was determined. It seems some of the treatment recommendation was not in accordance with NCCN guideline (e.g. patient number 2 with favorable intermediate cancer was offered with RT + ADT but RT only is adequate per NCCN guideline). Was the decision not guideline based?

Minor comments

3. Line 83 - "Gleason grade 4" should be "Gleason score 4" because "Gleason Grade" implies Grade group. Same is true for line 99.

4. Line 125, "Eighteen" is redundant

Author Response

Reviewer 1

Open Review

English language and style

( ) English very difficult to understand/incomprehensible
( ) Extensive editing of English language and style required
( ) Moderate English changes required
(x) English language and style are fine/minor spell check required
( ) I don't feel qualified to judge about the English language and style

Yes

Can be improved

Must be improved

Not applicable

Does the introduction provide sufficient background and include all relevant references?

( )

(x)

( )

( )

Are all the cited references relevant to the research?

(x)

( )

( )

( )

Is the research design appropriate?

( )

(x)

( )

( )

Are the methods adequately described?

( )

( )

(x)

( )

Are the results clearly presented?

( )

( )

(x)

( )

Are the conclusions supported by the results?

( )

( )

(x)

( )

Comments and Suggestions for Authors

The manuscript explored the utility of adding Oncotype DX GPS in treatment decision making for NCCN intermediate risk prostate cancer. The following comments should be addressed before the manuscript is considered for publication. 

Major comments:

  1. 1. Figure 1, the authors showed that the addition of GPS changed NCCN risk stratification. Please elaborate in detail how GPS changed patient's risk group since this is not described in the NCCN guideline.

Thank you for this comment. The combined risk group (NCCN+GPS) provided by Genomic Health and is estimated based on the Genomic score (GPS) and the components of the NCCN risk group (Gleason score, PSA level, clinical Stage, percentage of positive cores). We specified “ in the combined risk was provided by genomic Health” in the figure 1 description.

  1. Table 2 described treatment decision based on risk stratification. It is intriguing to notice that for patients of the same risk group, different treatment regimens were offered (eg. patient 2 and 3). Please provide more information on how the treatment regimen was determined. It seems some of the treatment recommendation was not in accordance with NCCN guideline (e.g. patient number 2 with favorable intermediate cancer was offered with RT + ADT but RT only is adequate per NCCN guideline). Was the decision not guideline based?

Thank you for this important remark. Indeed, the treatment suggested was not always in accordance with the NCCN guidelines.  The radiation therapy modality based on individual clinical parameters (PSA level, clinical stage, percentage of positive biopsies,age, NCCN risk group) with and without of the knowledge of GPS/ combined clinical and genomic risk group was left to the appreciation of the comity of radiation oncologists. Different therapies may be proposed for 2 patients within a same NCCN risk group. We believe one-size-fits-all treatment algorithm on the basis of risk classification alone might not be the most appropriate therapeutic approach.

This point represents an important limitation of the study and was added in the discussion section:

” Second, recommandations related to ADT and radiation volumes were left to the appreciation of the comity of radiation oncologists based on clinical parameters (PSA level, clinical stage, percentage of positive biopsies, NCCN risk group) and genomic score. Different therapies may be offered for 2 patients within a same NCCN risk group and some of the treatment suggestions were not in accordance with the NCCN guidelines. ”  line 253-257”.

Minor comments

  1. Line 83 - "Gleason grade 4" should be "Gleason score 4" because "Gleason Grade" implies Grade group. Same is true for line 99.

Thank you. These inaccuracies were corrected. Line 101 and 122.

  1. Line 125, "Eighteen" is redundant

Thank you. The redundancy was corrected. Line 148.

Reviewer 2 Report

General comment

The manuscript entitled “Genomic prostate score: a new tool to assess prognosis and optimize radiation therapy volumes and hormone therapy in intermediate-risk prostate cancer” aims to evaluate the ability of GPS in changing the therapeutic decision-making in intermediate-risk prostate cancer patients. The manuscript is overall fairly well written and the prospective nature of the work, together with the topic, which is among the most trending in the urological panorama, represents two points of strength. Nevertheless, the paper could be improved in some parts, starting from the lack of appropriate references (only 9 in all the work) and the scarce clarity of materials and methods. The suggested corrections are reported, in detail, below.

INTRODUCTION

The introduction could be expanded briefly reporting the epidemiology of prostate cancer and the impact of intermediate-risk prostate cancer.

51-53: add references. To this regard please also see: DOI: 10.3390/cancers14133272

MATERIALS AND METHODS

This section should be improved in terms of clarity in order to permit the reproducibility of the study. Add as much related information as available.

63: Add the institution and number of the approval.

RESULTS

In table 2, how did you obtain the GSP estimated risks at 10 years?

DISCUSSION

The discussion is poor and should be extended, evaluating similar studies reported in the literature, and adding the limitations of the study and future perspectives. This should be the main concern to correct.

Author Response

Reviewer 2

Open Review

English language and style

( ) English very difficult to understand/incomprehensible
( ) Extensive editing of English language and style required
( ) Moderate English changes required
(x) English language and style are fine/minor spell check required
( ) I don't feel qualified to judge about the English language and style

Yes

Can be improved

Must be improved

Not applicable

Does the introduction provide sufficient background and include all relevant references?

( )

(x)

( )

( )

Are all the cited references relevant to the research?

( )

( )

(x)

( )

Is the research design appropriate?

(x)

( )

( )

( )

Are the methods adequately described?

( )

(x)

( )

( )

Are the results clearly presented?

(x)

( )

( )

( )

Are the conclusions supported by the results?

(x)

( )

( )

( )

Comments and Suggestions for Authors

General comment

The manuscript entitled “Genomic prostate score: a new tool to assess prognosis and optimize radiation therapy volumes and hormone therapy in intermediate-risk prostate cancer” aims to evaluate the ability of GPS in changing the therapeutic decision-making in intermediate-risk prostate cancer patients. The manuscript is overall fairly well written and the prospective nature of the work, together with the topic, which is among the most trending in the urological panorama, represents two points of strength. Nevertheless, the paper could be improved in some parts, starting from the lack of appropriate references (only 9 in all the work) and the scarce clarity of materials and methods. The suggested corrections are reported, in detail, below.

INTRODUCTION

The introduction could be expanded briefly reporting the epidemiology of prostate cancer and the impact of intermediate-risk prostate cancer.

Thank you. The introduction was expanded including the epidemiology and the impact of intermediate-risk prostate cancer as suggested.

 “Prostate cancer (PCa) is the second most commonly diagnosed cancer and the sixth leading cause of cancer death among men worldwide, with an estimated 1 276 000 new cancer cases and 359 000 deaths in 2018. The global PCa burden is expected to rise to almost 2.3 million new cases and 740 000 deaths by 2040 simply due to the growth and aging of the population.” Line 34-38.

“In a review that focused on intermediate risk PCa, Zumsteg and Zelefsky presented this entity as a highly heterogenous one. This clinical heterogeneity makes application of a uniform treatment standard difficult especially knowing the fact more than 50% of variability in cure rates for PCa is related to clinical factors other than tumour stage, PSA level and Gleason score such as proportion of positive biopsy cores, perineural invasion, pretreatment PSA velocity and primary Gleason pattern”. Line 50-55.

51-53: add references. To this regard please also see: DOI: 10.3390/cancers14133272

Thank you. This reference was added as reference number 8.

MATERIALS AND METHODS

This section should be improved in terms of clarity in order to permit the reproducibility of the study. Add as much related information as available.

We agree with you. We tried to improve the clarity of this section and add some more information. For example, we added this paragraph in the material and methods section. line 97-102. “Formalin-fixed paraffin-embedded specimens, obtained from biopsy tissue were re-viewed by the pathologist according to the 2016 International Society of Urological Pathology consensus guidelines. Tissues were selected for analysis with the GPS assay from the biopsy core associated with the largest tumor amount with Gleason score 4. Eight unstained slides of 5µm were retrieved from the most predominant core biopsy with Gleason score 4.”

63: Add the institution and number of the approval.

Thank you. We provided the information requested.

“The study was conducted in accordance with the Declaration of Helsinki, and was ap-proved by the HOPITAL DE MONDOR Institutional Review Board (project number 16169, N° ID-RCB: 2016-A00789-42, 22/09/2016)”. Line 92-94.

RESULTS

In table 2, how did you obtain the GSP estimated risks at 10 years?

Thank you for this remark. GPS estimated risks at 10 years were based on GPS clinical validation study based on radical prostatectomy and .

DISCUSSION

The discussion is poor and should be extended, evaluating similar studies reported in the literature, and adding the limitations of the study and future perspectives. This should be the main concern to correct.

We agree with you. The discussion was almost completely rewritten with more references, limitation and futures perspectives.

Round 2

Reviewer 1 Report

Thank you for addressing the comments. No additional comments. 

Reviewer 2 Report

The authors improved the manuscript accordingly to previous suggestions. No further corrections are required.